# Gravity = Yang–Mills

**Roberto Bonezzi, Christoph Chiaffrino, Felipe Díaz-Jaramillo and Olaf Hohm ***

Institute for Physics, Humboldt University Berlin, Zum Großen Windkanal 6, D-12489 Berlin, Germany;
roberto.bonezzi@physik.hu-berlin.de (R.B.); chiaffrc@hu-berlin.de (C.C.); felipe.diaz-jaramillo@hu-berlin.de (F.D.-J.)
* Correspondence: ohohm@physik.hu-berlin.de

**Abstract:** This essay's title is justified by discussing a class of Yang–Mills-type theories of which standard Yang–Mills theories are special cases but which is broad enough to include gravity as a double field theory. We use the framework of homotopy algebras, where conventional Yang–Mills theory is the tensor product $\mathcal{K} \otimes \mathfrak{g}$ of a 'kinematic' algebra $\mathcal{K}$ with a color Lie algebra $\mathfrak{g}$. The larger class of Yang–Mills-type theories are given by the tensor product of $\mathcal{K}$ with more general Lie-type algebras, of which $\mathcal{K}$ itself is an example, up to anomalies that can be canceled for the tensor product with a second copy $\bar{\mathcal{K}}$. Gravity is then given by $\mathcal{K} \otimes \bar{\mathcal{K}}$.

**Keywords:** double copy; homotopy algebras; double field theory

The title of this essay is of course meant as a provocation, a provocation of the kind people engage in when they say such things as ER = EPR (denoting the conjectured correspondence [1] between Einstein–Rosen bridges (ER) [2] and Einstein–Podolsky–Rosen (EPR) entanglement [3]). Such statements appear nonsensical but perhaps suggest a novel interpretation of the terms involved that could be quite revealing. To say that gravity is a Yang–Mills theory is not nonsense but actually false if by Yang–Mills theory one means the standard text book theories labeled by the structure constants $f_{abc}$ of a 'color' Lie algebra $\mathfrak{g}$ and by gravity one means the Einstein–Hilbert action in the same dimension. Indeed, in the context of perturbative quantum field theory (QFT), Yang–Mills theories carry only cubic and quartic couplings and are renormalizable, while Einstein–Hilbert gravity is non-polynomial and non-renormalizable.

Rather, here, we suggest the definition of a broader class of 'Yang–Mills-type' theories based on the homotopy algebra formulation of field theories to be explained below [4–6]. In this framework, the textbook Yang–Mills theory takes the form of a tensor product $\mathcal{K} \otimes \mathfrak{g}$ of a homotopy algebra $\mathcal{K}$ that encodes the kinematics of Yang–Mills theory, with the color Lie algebra $\mathfrak{g}$ [7]. One may then define the larger class of Yang–Mills-type theories as the tensor product of $\mathcal{K}$ with more general 'Lie-type' algebras, namely homotopy Lie or $L_\infty$ algebras. The kinematic algebra $\mathcal{K}$ itself carries, in a hidden way, such a Lie-type algebra up to obstructions that, however, cancel on a subspace of the tensor product $\mathcal{K} \otimes \bar{\mathcal{K}}$ with a second copy $\bar{\mathcal{K}}$, as has been proved to the order relevant for quartic couplings [8–10]. We may thus define the yet larger class of Yang–Mills-type theories as those given by the tensor product of $\mathcal{K}$ with an obstructed Lie-type algebra for which the obstructions are canceled on a non-trivial subspace. So defined, it is just a fact that there is a subspace of $\mathcal{K} \otimes \bar{\mathcal{K}}$ defining a Yang–Mills-type theory that *is* gravity. Importantly, this theory is *not* Einstein–Hilbert gravity but rather double field theory, which includes the graviton, an antisymmetric tensor (B-field) and a scalar (dilaton) [11–13].

Our construction is directly inspired by, and a gauge invariant off-shell generalization of, the 'double copy' technique of amplitudes [14–16], known to the general public under the slogan 'Gravity = (Yang–Mills)$^2$'. We think that the slogan 'Gravity = Yang–Mills' is more appropriate, a viewpoint that can perhaps be justified most succinctly as follows: Since the standard textbook presentation of Yang–Mills theory gives a Lagrangian in terms of structure constants $f_{abc}$, we can think of Yang–Mills theory as a machine: a machine that

takes as input structure constants $f_{abc}$ and produces as output a QFT. The framework of homotopy algebra allows us to reinvent Yang–Mills theory as a more powerful machine that accepts as input more general 'Lie-type' algebras. Feeding in a conventional Lie algebra, this machine produces standard Yang–Mills theory, but feeding in a second copy of the kinematic algebra itself produces double field theory.

Let us begin by explaining the relationship between homotopy algebras and field theories. A field theory is defined by introducing a set of fields, which here we denote collectively by $\psi$, by specifying field equations or an action and, possibly, by specifying gauge symmetries and their dual Noether/Bianchi identities. We will focus on perturbation theory, where the fields form a vector space (the sum of two fields is again a field), and so do the gauge parameters, etc. More precisely, the totality of these objects form a *graded* vector space $X$, which means that we assign, as a book-keeping device, an integer called degree to each object, depending, e.g., on whether it is a field or a gauge parameter. Such gradings are familiar from the BV-BRST formalism, where they are related to the ghost number [17,18].

One then defines various maps on this graded vector space in order to encode, for instance, kinetic terms, interaction vertices, and gauge transformations, which define the theory. Specifically, at least in perturbation theory, one can write the action as

$$S = \frac{1}{2}\langle \psi, B_1(\psi) \rangle + \frac{1}{3!}\langle \psi, B_2(\psi, \psi) \rangle + \frac{1}{4!}\langle \psi, B_3(\psi, \psi, \psi) \rangle + \cdots , \tag{1}$$

i.e., as a sum of quadratic terms, cubic terms, and so on. We have written the terms of order $n + 1$ in fields using a universal inner product $\langle \, , \, \rangle$ and multilinear maps $B_n$ with $n$ arguments. Furthermore, one assumes, for instance, that the field equations and gauge transformations with gauge parameters $\lambda$, respectively, are given by

$$B_1(\psi) + \frac{1}{2}B_2(\psi, \psi) + \frac{1}{3!}B_3(\psi, \psi, \psi) + \cdots = 0 , \tag{2}$$

and

$$\delta\psi = B_1(\lambda) + B_2(\lambda, \psi) + \frac{1}{2}B_3(\lambda, \psi, \psi) + \cdots . \tag{3}$$

It must be emphasized that the $B_n$, when evaluated on different objects such as fields or gauge parameters, are a priori independent maps, distinguished by the degrees of their inputs. For instance, $B_1$, which is known as the differential, acts according to the following chain:

$$\cdots \longrightarrow X_{-1} \xrightarrow{B_1} X_0 \xrightarrow{B_1} X_1 \xrightarrow{B_1} X_2 \longrightarrow \cdots \tag{4}$$

$$\lambda \qquad \psi \qquad \text{EoM} \qquad \text{Noether}$$

where the interpretation of each space is indicated in the second line. According to (3), $B_1$ acting on $\lambda \in X_{-1}$ is defined by the zeroth-order, inhomogeneous terms of the gauge transformations, while according to (2), $B_1$ acting on $\psi \in X_0$ is defined by the linear terms of the equations of motion. For instance, in Yang–Mills theory, $B_1$ on fields is the second-order differential 'Maxwell operator', while $B_1$ on gauge parameters is the first-order differential operator of the Abelian gauge transformation. Linearized gauge invariance requires

$$\delta(B_1(\psi)) = B_1(B_1(\lambda)) = 0 . \tag{5}$$

This relationship is summarized as $B_1^2 = 0$, which moreover must hold on the entire complex (4), where it also encodes linearized Noether identities.

More generally, the $B_n$ are determined so that the non-linear equations of motion, gauge transformations, and so on correctly describe the desired field theory. They must, of course, satisfy consistency conditions following from those of field theory, such as, for

instance, gauge covariance of the field equations or closure of the gauge algebra. Thus, with any consistent (perturbative) field theory, we are given a graded vector space $X$ equipped with maps $B_n$ obeying various consistency relations. This is what mathematicians call a *structure*. So what is the general structure of field theory? This is the category of homotopy Lie algebras or $L_\infty$ algebras. These generalizations of Lie algebras are defined as integer graded vector spaces $X = \bigoplus_{i \in \mathbb{Z}} X_i$ equipped with multilinear maps $B_n$ of intrinsic degree $+1$ (i.e., the degree of the output is the sum of the input degrees plus 1), which are graded symmetric (e.g., $B_2(x_1, x_2) = (-1)^{x_1 x_2} B_2(x_2, x_1)$, where in exponents, $x$ denotes the degree). This means that, depending on the input, the map may be symmetric, as when evaluated on fields, or antisymmetric, as when evaluated on gauge parameters. The $B_n$ are subject to an infinite number of $L_\infty$ relations. The first relation is $B_1^2 = 0$, and we display the next two relations:

$$B_1(B_2(x_1, x_2)) + B_2(B_1(x_1), x_2) + (-1)^{x_1} B_2(x_1, B_1(x_2)) = 0 \,,$$

$$\begin{aligned} &B_2(B_2(x_1, x_2), x_3) + (-1)^{x_3(x_1+x_2)} B_2(B_2(x_3, x_1), x_2) + (-1)^{x_1(x_2+x_3)} B_2(B_2(x_2, x_3), x_1) \\ &+ B_1(B_3(x_1, x_2, x_3)) + B_3(B_1(x_1), x_2, x_3) + \text{two terms} = 0 \,. \end{aligned} \quad (6)$$

These state, respectively, that the differential $B_1$ obeys the Leibniz rule with respect to the 2-bracket $B_2$ and that the Jacobi identity only needs to hold 'up to homotopy', with the failure being governed by the differential $B_1$ and the '3-bracket' $B_3$. The $L_\infty$ relations encode the consistency conditions of field theory, for instance, the gauge covariance of (2) under (3).

We now turn to the $L_\infty$ algebra of Yang–Mills theory. The familiar action, written in terms of the non-Abelian field strength $F_{\mu\nu}{}^a = \partial_\mu A_\nu{}^a - \partial_\nu A_\mu{}^a + f^a{}_{bc} A_\mu{}^b A_\nu{}^c$, reads

$$S_{\text{YM}} = -\frac{1}{4} \int dx \, F^{\mu\nu a} F_{\mu\nu a} \,, \quad (7)$$

where $dx$ denotes the flat space volume element in $D$ dimensions. Expanded in fields and using integrations by part, this becomes

$$S_{\text{YM}} = \int dx \left[ \tfrac{1}{2} A_a^\mu \Box A_\mu^a + \tfrac{1}{2} (\partial^\mu A_\mu^a)^2 - f_{abc} \, \partial_\mu A_\nu^a A^{\mu b} A^{\nu c} - \tfrac{1}{4} f^e{}_{ab} f_{ecd} A_\mu^a A_\nu^b A^{\mu c} A^{\nu d} \right] \,, \quad (8)$$

where $\Box = \partial^\mu \partial_\mu$ denotes the d'Alembert operator. For the following applications, it is useful to pass to an equivalent formulation by introducing an auxiliary scalar:

$$S_{\text{YM}} = \int dx \left[ \tfrac{1}{2} A_a^\mu \Box A_\mu^a - \tfrac{1}{2} \varphi_a \varphi^a + \varphi_a \, \partial^\mu A_\mu^a + \cdots \right] \,, \quad (9)$$

with the ellipsis denoting the same cubic and quartic terms as in (8). Integrating out $\varphi_a$, one recovers (8).

One could now determine the $L_\infty$ algebra of Yang–Mills theory as sketched above, but it is more useful to immediately 'strip off' color and to write this algebra as a tensor product of the color Lie algebra $\mathfrak{g}$ with another kind of homotopy algebra. The latter 'kinematic' algebra is defined on a vector space $\mathcal{K}$ that carries the same objects as the $L_\infty$ algebra of Yang–Mills but without color indices, which by an abuse of language we denote by the same letters and names. $\mathcal{K}$ defines a chain complex (with degrees shifted by one)

$$K_0 \xrightarrow{\ m_1\ } K_1 \xrightarrow{\ m_1\ } K_2 \xrightarrow{\ m_1\ } K_3$$

$$\begin{array}{ll} \mathcal{K}^{(0)}: & \lambda \quad\;\; A_\mu \quad\;\; E \\[2pt] & \qquad {\scriptstyle b}\quad\quad {\scriptstyle b}\quad\quad {\scriptstyle b} \\ \mathcal{K}^{(1)}: & \qquad\;\; \varphi \quad\;\; E_\mu \quad\;\; \mathcal{N} \end{array} \quad (10)$$

where the differential $m_1$ satisfies $m_1^2 = 0$. For instance, on gauge parameters and fields, respectively, $m_1$ acts as

$$m_1(\lambda) = \begin{pmatrix} \partial_\mu \lambda \\ \Box \lambda \end{pmatrix} \in K_1 \,, \qquad m_1 \begin{pmatrix} A_\mu \\ \varphi \end{pmatrix} = \begin{pmatrix} \partial \cdot A - \varphi \\ \Box A_\mu - \partial_\mu \varphi \end{pmatrix} \in K_2 \,, \tag{11}$$

where we use the short-hand notation $\partial \cdot A = \partial_\nu A^\nu$. Thanks to the introduction of $\varphi$, we have a $\mathbb{Z}_2$ grading $\mathcal{K} = \mathcal{K}^{(0)} \oplus \mathcal{K}^{(1)}$ and a map $b$ of intrinsic degree $-1$, whose action is indicated in the diagram (10). For instance, $b(A_\mu, \varphi) = \varphi$, where the output is re-interpreted as a gauge parameter and hence degree shifted. In addition, $\mathcal{K}$ carries a graded symmetric 2-product $m_2$ of degree zero and a 3-product $m_3$ of degree $-1$, which we display evaluated on the fields

$$m_2^\mu(A_1, A_2) = \partial \cdot A_1 A_2^\mu + 2 A_1 \cdot \partial A_2^\mu + \partial^\mu A_1 \cdot A_2 - (1 \leftrightarrow 2) \,, \tag{12}$$
$$m_3^\mu(A_1, A_2, A_3) = A_1 \cdot A_2 A_3^\mu + A_3 \cdot A_2 A_1^\mu - 2 A_1 \cdot A_3 A_2^\mu \,,$$

where the external $\mu$ index indicates the vector component in $K_2$.

A graded vector space $\mathcal{K}$ equipped with maps $m_1, m_2, m_3$ and possibly higher maps, subject to certain symmetry properties, is called a $C_\infty$ algebra (the homotopy version of a commutative associative algebra) provided that, in addition to $m_1^2 = 0$, the following relations hold:

$$m_1(m_2(u_1, u_2)) = m_2(m_1(u_1), u_2) + (-1)^{u_1} m_2(u_1, m_1(u_2)) \,,$$
$$m_2(m_2(u_1, u_2), u_3) - m_2(u_1, m_2(u_2, u_3)) = m_1(m_3(u_1, u_2, u_3)) + m_3(m_1(u_1), u_2, u_3) \tag{13}$$
$$+ (-1)^{u_1} m_3(u_1, m_1(u_2), u_3) + (-1)^{u_1 + u_2} m_3(u_1, u_2, m_1(u_3)) \,.$$

The first relation is the Leibniz rule. The second relation states that $m_2$ is associative 'up to homotopy', and in general, there may be infinitely many more relations. A $C_\infty$ algebra is a special case of an $A_\infty$ algebra where the $m_2$ is graded symmetric, while the higher $m_n$ for $n \geqslant 3$ are subject to graded Young–Tableaux-type symmetries.

The $L_\infty$ algebra of Yang–Mills theory is now obtained from the $C_\infty$ algebra $\mathcal{K}$ by tensoring with the color Lie algebra $\mathfrak{g}$ [7] (see [19] for a review):

$$X_{\mathrm{YM}} = \mathcal{K} \otimes \mathfrak{g} \,. \tag{14}$$

At the level of the vector space, this just means that the objects in (10) are made $\mathfrak{g}$-valued by decorating them with color indices (and degree shifting by one):

$$x = u^a \otimes t_a \in X_{\mathrm{YM}} \,, \tag{15}$$

where $t_a$ are the generators of $\mathfrak{g}$. One obtains the fields, gauge parameters, etc., of Yang–Mills theory. The $B_n$ encoding the $L_\infty$ structure on (14) are

$$B_1(x) = m_1(u^a) \otimes t_a \,,$$
$$B_2(x_1, x_2) = (-1)^{x_1} m_2(u_1^a, u_2^b) f_{ab}{}^c \otimes t_c \,, \tag{16}$$
$$B_3(x_1, x_2, x_3) = \left[ (-1)^{x_2} m_3(u_1^a, u_2^b, u_3^c) + (-1)^{x_1(x_2+1)} m_3(u_2^a, u_1^b, u_3^c) \right] f_{ad}{}^e f_{bc}{}^d \otimes t_e \,,$$

where in this case all $B_n$ for $n > 3$ are zero. The $L_\infty$ relations on $X_{\mathrm{YM}}$ follow from the $C_\infty$ relations on $\mathcal{K}$ and the Jacobi identities for $f_{ab}{}^c$. The resulting $B_2$ and $B_3$ encode the full Yang–Mills theory; in particular, using (12), they reproduce the cubic and quartic interactions.

With the decomposition (14) of homotopy algebras, we have separated Yang–Mills theory into its 'kinematic' and its 'color' parts, with the former being an 'associative-type' algebra and the latter a 'Lie-type' algebra. The idea inspired by the double copy procedure of amplitudes is to replace $\mathfrak{g}$ in (14) by another type of Lie algebra based on the kinematics

of Yang–Mills theory ('kinematic Lie algebra') in order to obtain gravity. While $\mathcal{K}$ started its life as an associative-type algebra, it actually also admits a hidden Lie-type algebra (in a suitably generalized sense). Borrowing terminology from linguistics, we may refer to the former as the 'surface structure' and the latter as the 'deep structure'.

In order to display this deep structure, we use the map $b$ defined in (10), which is nilpotent, $b^2 = 0$, to define a new Lie-type bracket on $\mathcal{K}$ as the failure of $b$ to act via the Leibniz rule on $m_2$:

$$b_2(u_1, u_2) := bm_2(u_1, u_2) - m_2(bu_1, u_2) - (-1)^{u_1} m_2(u_1, bu_2) .\qquad(17)$$

From this definition, it follows with $b^2 = 0$ that $b$ obeys the Leibniz rule with $b_2$, and hence, $b$ can be thought of as a second differential of opposite degree to $m_1$. The deep structure on $\mathcal{K}$ is a generalization of a Batalin–Vilkovisky (BV) algebra. A BV algebra consists of a (graded) commutative and associative product together with a nilpotent operator that, however, does not act via the Leibniz rule on the product but rather is of 'second order' (similar to the BV Laplacian of the BV formalism). Defining then a 2-bracket as the failure of the differential to act via the Leibniz rule on the product, one obtains a Lie bracket satisfying the Jacobi identities and a compatibility condition with the product. The differential $b$, the 2-product $m_2$, and the 2-bracket $b_2$ above want to be a BV algebra but fail to be that because (i) $b$ is not second order with respect to $m_2$, and (ii) $m_2$ is not associative. These failures suggest that there is a homotopy BV algebra (BV$_\infty$ algebra [20]), but there is an additional failure due to the relation

$$m_1 b + b m_1 = \square ,\qquad(18)$$

where $\square$ denotes the d'Alembert operator. This relation quickly follows with (10) and (11), and it means that $m_1$ and $b$ are compatible only up to '$\square$–failures'. This has various ramifications. For instance, the original differential $m_1$ does not obey the Leibniz rule with respect to $b_2$:

$$m_1(b_2(u_1, u_2)) + b_2(m_1(u_1), u_2) + (-1)^{u_1} b_2(u_1, m_1(u_2)) = 2 \, m_2(\partial^\mu u_1, \partial_\mu u_2) ,\qquad(19)$$

where the 'anomaly' on the right-hand side follows from (17), (18), and $\square$ being second-order. Similar $\square$-failures appear in other relations. Formalizing these failures one can define a more general algebraic structure that is realized on the kinematic space $\mathcal{K}$ of Yang–Mills theory, which following Reiterer we denote by BV$_\infty^\square$ [21]. This includes as a subalgebra a $C_\infty$ algebra and as a 'subsector' an $L_\infty$ algebra that, however, is obstructed by $\square$-failures. (An operator $b$ and an associated BV$_\infty^\square$ algebra are also realized in self-dual Yang–Mills theory [22] and 3D Chern–Simons theory [8,23,24].)

We can now turn to the construction of gravity in the form of double field theory (DFT). Due to the $\square$-failures, a general BV$_\infty^\square$ algebra is not quite of 'Lie-type' and cannot be tensored with the kinematic algebra $\mathcal{K}$ to obtain a genuine $L_\infty$ algebra of gravity. However, taking a second copy $\bar{\mathcal{K}}$ of the kinematic algebra itself, these failures can be canceled on a subspace of the tensor product $\mathcal{K} \otimes \bar{\mathcal{K}}$. Denoting all objects of $\bar{\mathcal{K}}$ with a bar, the full tensor product space $\mathcal{K} \otimes \bar{\mathcal{K}}$ is a chain complex carrying two natural differentials of opposite degrees:

$$B_1 := m_1 \otimes \mathbf{1} + \mathbf{1} \otimes \bar{m}_1 , \qquad b^- := \tfrac{1}{2}(b \otimes \mathbf{1} - \mathbf{1} \otimes \bar{b}) ,\qquad(20)$$

which both square to zero due to $m_1^2 = b^2 = 0$. The $\square$-failure relation (18) now implies

$$B_1 b^- + b^- B_1 = \Delta , \qquad \Delta := \tfrac{1}{2}(\square - \bar{\square}) .\qquad(21)$$

We can eliminate the '$\Delta$-failure' by going to a subspace with $\Delta = 0$. To explain this point, we first note that since $\mathcal{K}$ is a space of functions of coordinates $x$, and $\bar{\mathcal{K}}$ is a space of functions of coordinates $\bar{x}$, $\mathcal{K} \otimes \bar{\mathcal{K}}$ is a space of functions of doubled coordinates $(x, \bar{x})$. (This is familiar from quantum mechanics: the tensor product of two one-particle Hilbert

spaces of wave functions of one coordinate yields the two-particle Hilbert space of wave functions of two coordinates). We may then impose $\Delta = 0$ on functions and products of functions, which in DFT is known as the strong constraint and essentially equivalent to identifying coordinates $x$ with coordinates $\bar{x}$. We will return to the 'weakly constrained' case momentarily.

We thus consider the subspace

$$\mathcal{V}_{\mathrm{DFT}} := \left\{ \psi \in \mathcal{K} \otimes \bar{\mathcal{K}} \,\middle|\, \Delta \psi = 0 \,,\, b^- \psi = 0 \right\} \,, \tag{22}$$

where $b^- \psi = 0$ restricts the spectrum appropriately. Together, both constraints in here are known as level-matching constraints. The space $\mathcal{V}_{\mathrm{DFT}}$ is precisely the complex of DFT as derived from closed string field theory in [11]. For instance, the fields in degree zero (with an overall degree shift by 2) are given by

$$(e_{\mu\bar{\nu}} \,,\, e \,,\, \bar{e} \,,\, f_\mu \,,\, \bar{f}_{\bar{\mu}}) \,\in\, (K_1 \otimes \bar{K}_1) \oplus (K_0 \otimes \bar{K}_2) \oplus (K_2 \otimes \bar{K}_0) \,, \tag{23}$$

where $e_{\mu\bar{\nu}}$ encodes spin-2 and B-field fluctuations, $e$ and $\bar{e}$ are two 'dilatons', one of which is pure gauge, and $f_\mu$ and $\bar{f}_{\bar{\mu}}$ are auxiliary fields that can be integrated out. More generally, $\mathcal{V}_{\mathrm{DFT}}$ encodes the gauge parameters and gauge-for-gauge parameters of DFT, etc., so that, for instance, $\delta\psi = B_1(\Lambda)$ implies $\delta e_{\mu\bar{\nu}} = \partial_\mu \bar{\lambda}_{\bar{\nu}} + \bar{\partial}_{\bar{\nu}} \lambda_\mu$, exhibiting the 'double copy' structure of linearized diffeomorphisms and B-field gauge transformations.

Turning to the non-linear structure, we have to define higher brackets on $\mathcal{V}_{\mathrm{DFT}}$. The 2-bracket can be written, in an input-free notation explained in [8], as

$$B_2 := -\tfrac{1}{4} \left( b_2 \otimes \bar{m}_2 - m_2 \otimes \bar{b}_2 \right) = -\tfrac{1}{2} \, b^- (m_2 \otimes \bar{m}_2) \,, \tag{24}$$

where the second equality holds on the subspace $b^- = 0$. This 2-bracket obeys the Leibniz relation, c.f. (6), thanks to $b^-$ anticommuting with $B_1$ for $\Delta = 0$. The second $L_\infty$ relation in (6) involving the Jacobiator can be satisfied upon defining a suitable $B_3$, which is more involved but can be written entirely in terms of the $\mathrm{BV}_\infty^\square$ structures of Yang–Mills theory [8]. With the above general $L_\infty$ dictionary, this determines the complete gravity theory to quartic order, in particular, via (1), the quartic couplings.

Let us emphasize two features of this construction of gravity as a Yang–Mills-like theory:

- The gauge algebra of DFT, which is a duality covariant version of the diffeomorphism algebra of gravity, originates rather directly from the couplings of Yang–Mills theory.
- 4-graviton amplitudes can be computed with the $B_2$ and $B_3$ above and by construction exhibit the factorization into Yang–Mills amplitudes.

Above, we have essentially identified the two coordinates $x$ and $\bar{x}$ in order to obtain '$\mathcal{N} = 0$ supergravity' (Einstein gravity coupled to B-field and dilaton) as a strongly constrained DFT. It is, however, possible to obtain a weakly constrained DFT, at least if all dimensions are toroidal, in which the fields genuinely depend on $x$ and $\bar{x}$, subject only to $\square = \bar{\square}$ (level-matching for string theory on tori). One uses that the total space $\mathcal{K} \otimes \bar{\mathcal{K}}$ carries a $\mathrm{BV}_\infty^\Delta$ structure and performs an operation known as homotopy transfer (see, e.g., [25–28]) to the subspace $\Delta = 0$, together with an additional non-local shift of $B_3$ [29]. The resulting space realizes an $L_\infty$ algebra of weakly constrained functions and hence defines a consistent field theory containing momentum and winding modes without other higher string modes. The momentum and winding modes are encoded in the doubled coordinate dependence of the massless fields. Hull and Zwiebach constructed such a theory to cubic order starting from string field theory in 2009 [11], but since then, it has been an open problem to construct the theory to quartic and higher orders.

We close this essay with a tantalizing possibility. Suppose a weakly constrained DFT with doubled compact coordinates and standard non-compact coordinates can be constructed to all orders in fields. This theory is expected to have an improved UV behavior.

While the strongly constrained theory must exhibit the usual UV divergencies of general relativity, a weakly constrained theory features infinite towers of additional massive states (that in particular are charged under diffeomorphisms along the non-compact dimensions). Since these states run in loops, this theory should have an improved UV behavior. Indeed, as shown by Sen [30], a weakly constrained DFT can in principle be derived from the full closed string field theory upon integrating out all string modes that are not part of the DFT sector [28]. The theory so constructed then inherits the UV finiteness of the full string theory [30]. Therefore, constructing a weakly constrained DFT from scratch might lead, possibly upon including $\alpha'$ corrections [31,32], to a consistent theory of quantum gravity. Perhaps there is a quantum theory of gravity much 'smaller' than the currently explored string theories, and perhaps this quantum gravity is secretly a Yang–Mills theory.

　　　*Note added:* After submitting this essay to the arxiv, we were informed by Anton Zeitlin that his early papers [7,33] already contain $BV_\infty$ structures, which, remarkably, have even been suggested to relate to gravity along lines closely related to the above discussion [34].

**Author Contributions:** Conceptualization, R.B., C.C., F.D.-J. and O.H.; validation, R.B. and F.D.-J.; writing—original draft preparation, O.H.; writing—review and editing, R.B., C.C. and F.D.-J.; supervision, O.H.; project administration, O.H.; funding acquisition, O.H. All authors have read and agreed to the published version of the manuscript.

**Funding:** This work is funded by the European Research Council (ERC) under the European Union's Horizon 2020 research and innovation programme (grant agreement No 771862) and by the Deutsche Forschungsgemeinschaft (DFG, German Research Foundation), "Rethinking Quantum Field Theory", project number 417533893/GRK2575.

**Data Availability Statement:** Data sharing not applicable.

**Acknowledgments:** We thank Barton Zwiebach for his discussions and Anton Zeitlin for his correspondence.

**Conflicts of Interest:** The authors declare no conflict of interest.

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
