# Peer review of "Gravity = Yang–Mills"

_symmetry, doi:10.3390/sym15112062_

Round 1
Reviewer 1 Report
Comments and Suggestions for Authors
The paper is well written, with some new results. The observation that in the double field theory some operators have additional properties not present in the single field theory is interesting. I think the `essay' is publishable. However I suggest some improvements
(i) The new observations are not stated explicitly in the abstract and require to be highlighted in the text of the paper.
(ii) E-H action in linearized approximation is polynomial. If the authors mean something else apart from the non-polynomial measure in the E-H action in the statement in Page 2, kindly clarify.
(iii) Perhaps the open problem of the Hull and Zweibach reference [8] should be elaborated upon.
Author Response
We thank the referee for the positive remarks and constructive suggestions
for improvement. In response to these we had adapted the text as follows:
(i) We have added some clarification in the first paragraph on p. 11 regarding
the additional states of double field theory once the latter is taken to be
"weakly constrained". In this case double field theory encodes, in addition to
the familiar massless states, massive momentum and winding modes,
which are implicitly encoded in the dependence of the "massless" fields
in the doubled internal coordinates.
(ii) We agree that the E-H action to linearized order is quadratic and hence
polynomial in fields, but what we are referring to here is the complete expansion
of the E-H action, including cubic and all higher orders, in which case the action
is non-polynomial. We hope that the reference to "perturbative QFT" in that
paragraph makes this clear.
(iii) We have changed and expanded the last sentence of the first paragraph
on p. 11 in order to say in more detail what the results of Hull and Zwiebach were
and in which case an a problem is solved that was open before.
Reviewer 2 Report
Comments and Suggestions for Authors
The essay deals with a provocative proposal of an equivalence of gravity with the Yang-Mills theory. The work is intriguing and well written, and I have no specific objection to the publication. I just suggest to explicitly explain the acronyms (ER, EPR, QFT,...) when they are used for the first time.
Author Response
We thank the referee for the positive remarks, and we agree that an
explanation of acronyms is warranted, which we have done in the
updated version.